# New Sulfur Organic Polymer-Concrete Composites Containing Waste Materials: Mechanical Characteristics and Resistance to Biocorrosion

**DOI:** 10.3390/ma12162602

**Published:** 2019-08-15

**Authors:** Beata Gutarowska, Renata Kotynia, Dariusz Bieliński, Rafał Anyszka, Jakub Wręczycki, Małgorzata Piotrowska, Anna Koziróg, Joanna Berłowska, Piotr Dziugan

**Affiliations:** 1Institute of Technology Fermentation and Microbiology, Faculty of Biotechnology and Food Science, Lodz University of Technology, Wolczanska 171/173, 90-924 Lodz, Poland; 2Department of Concrete Structures, Faculty of Civil Engineering, Architecture and Environmental Engineering, Lodz University of Technology, Al. Politechniki 6, 90-924 Lodz, Poland; 3Institute of Polymer and Dye Technology, Faculty of Chemistry, Lodz University of Technology, Stefanowskiego 12/16, 90-924 Lodz, Poland

**Keywords:** sulfur-polymer concrete composites, biocorrosion

## Abstract

The aim of this study was to develop new sulfur-copolymer concrete composites using waste compounds that have good mechanical characteristics and show a resistance to biocorrosion. The comonomers used to synthesize the sulfur–organic copolymers were—90 wt. % sulfur; 5 wt. % dicyclopentadiene (DCPD); 5 wt. % organic monomers, styrene (SDS), 1-decene (SDD), turpentine (SDT), and furfural (SDF). The concrete composites based on sulfur–organic copolymers were filled with aggregates, sand, gravel, as well as additives and industrial waste such as fly ash or phosphogypsum. The sulfur–organic copolymers were found to be chemically stable (softening temperature, thermal stability, melting temperature, amount of recrystallized sulfur, and shore D hardness). Partial replacement of DCPD with other organic comonomers did not change the thermal stability markedly but did make the copolymers more elastic. However, the materials became significantly stiffer after repeated melting. All the tested copolymers were found to be resistant to microbial corrosion. The highest resistance was exhibited by the SDS-containing polymer, while the SDF polymer exhibited the greatest change due to the activity of the microorganisms (FTIR analysis and sulfur crystallization). The concrete composites with sulfur–organic copolymers containing DCPD, SDS, SDF, fly ash, and phosphogypsum were mechanically resistant to compression and stretching, had low water absorbance, and were resistant to factors, such as temperature and salt. Resistance to freezing and thawing (150 cycles) was not confirmed. The concrete composites with sulfur–organic copolymers showed resistance to bacterial growth and acid activity during 8 weeks of incubation with microorganisms. No significant structural changes were observed in the SDS composites after incubation with bacteria, whereas composites containing SDF showed slight changes (FTIR and microscopic analysis). The concrete composite containing sulfur, DCPD, SDS, sand, gravel, and fly ash was the most resistant to microbiological corrosion, based on the metabolic activity of the bacteria and the production of ergosterol by the molds after eight weeks of incubation. It was found that *Thiobacillus thioparus* was the first of the acidifying bacteria to colonize the sulfur concrete, decreasing the pH of the environment. The molds *Penicillium chrysogenum*, *Aspergillus versicolor* and *Cladosporium herbarum* were able to grow on the surface of the tested composites only in the presence of an organic carbon source (glucose). During incubation, they produced organic acids and acidified the environment. However, no morphological changes in the concretes were observed suggesting that sulfur–organic copolymers containing styrene could be used as engineering materials or be applied as binders in sulfur-concretes.

## 1. Introduction

Under ambient conditions, sulfur is a crystalline material of yellow color. The most common crystalline form of sulfur is composed of eight atom rings. When subjected to heating, sulfur undergoes spontaneous radical self-polymerization at 159 °C via decomposition of the crystalline rings. The addition of selenium to sulfur decreases the polymerization temperature. For example, the addition of 25% selenium results in a decrease in the polymerization temperature from 159 °C to ca. 122 °C [1]. The polymerization of sulfur can also be induced below 159 °C, by laser treatment [2,3]. However, such polymeric structures are energetically unstable and undergo relatively rapid recrystallization deteriorating the mechanical properties of the material. The polymeric structure of sulfur can be set using unsaturated organic compounds, which are radically copolymerized in order to stabilize the sulfur–organic copolymers that are created. The main comonomers that are applied for this purpose are—dicyclopentadiene, which produces a fragile material with a branched and cross-linked structure; styrene, which leads to an olygomeric paste composed of linear macromolecules; diynes, in the presence of which polytiophenes are created; disulphides, which also react with sulfur [4,5,6,7,8]. Crystalline sulfur also undergoes anionic polymerization with propylene sulphide, producing linear polysulphides. 

However, perhaps the most interesting comonomer for synthesis of sulfur–organic copolymers is 1,3-diizopropylobenzene [8]. Application of this comonomer results in the production of thermoplastic copolymers exhibiting high processability and tremendous potential for applications in the electrical industry as cathodes in lithium cells [9], or being utilized as camera lenses operating in the near and medium infrared ranges, in the optical industry [10,11]. The possible application of sulfur copolymers as materials for lithium cells was recognized at the beginning of the 21st century [12], opening a fast-growing utilization area. Sulfur copolymers can also be applied as sulfur donors, improving dispersion and practically eliminating the sulfur blooming in rubber vulcanizates. This makes them an interesting alternative for sulfur vulcanization in rubber applications, including tires (relatively polar sulfur exhibits limited solubility in unipolar rubber matrices). Research in this direction had already begun in earnest in the 1970s [13].

Sulfur–organic copolymers can also be applied in the field of civil engineering as an alternative binder for concretes. Sulfur concrete has a wide range of possible applications including road construction, structures exposed to aggressive chemical environments utilized in chemical, petroleum and food industries, and in farming [14,15,16,17,18,19,20]. For instance, Patent WO 2012/138860A1 [18] describes the composition of a bitumen mix made of 20–80 wt.% of bitumen, 0.1–7 wt.% of organic copolymer, and 20–60 wt.% of sulfur. Patent WO 2014/206986A1 [19] refers to asphalt with pelletized sulfur compounds containing calcium lignosulphonate. Patent WO 2015/014953A1 [20] covers the application of organosilanes with long-chain functionalities for the modification of sulfur composites.

The properties of sulfur concretes are provided by the sulfur polymer binder, which is produced in the process of ordinary sulfur synthesis with variable modifiers. However, initial results have shown that the new material has a tendency to lose its mechanical properties during prolonged exposure to thermal treatment or other energy transfer, due to degradation of the internal sulfur–polymer bonds [21]. To solve this problem, several modifiers with various properties have been investigated as sulfur binders [22,23]. A significant advantage of using a modified sulfur polymer binder for sulfur concrete production is that it can provide full compressive strength very rapidly (24 h after forming). According to the literature, sulfur concrete has high resistance to harsh acid and salt environments. It is also resistant to biocorrosion, radiation activity, and freeze–thaw cycles. It is hardly liquid-absorbing material, which is easy to recycle by melt re-forming [22,24]. However, one of the most important disadvantage of sulfur concrete is its low heat resistance limiting its applicability in civil engineering constructions. This is caused by a relatively low melting point temperature of sulfur—around 115 °C (239 °F). A number of sulfur concrete properties also depend primarily on the content of the sulfur binder in the concrete composition, changing significantly with an increase in the sulfur content. The most important properties of sulfur concretes limiting their application are flammability, linear coefficient of thermal expansion, and volume shrinkage during melt–solid solicitation, which can lead to crack formation in large volumes of sulfur concrete.

The aim of the present study was to develop new sulfur–copolymer concrete composites using waste compounds that have good mechanical properties and show a high resistance to biocorrosion. Due to increasing stockpiles, there is a great interest in finding new applications for the sulfur that remains after industrial processes [16,25]. According to GUS (the Central Statistical Office of Poland), 131.3 million tons of industrial waste materials were produced in Poland and only 27.6% of them was recycled [26]. The total mass of waste kept on stockpiles has reached 1,683.5 million tons in Poland. Slag and fly ash (products of coal combustion) are the most common types of stockpiles located close to power plants and electrical power and heating stations. Replacing cement with a sulfur polymer binder has been proposed as a way of utilizing these wastes, due to its high chemical tolerance. Most of such materials could be used to produce sulfur concretes but its possible applications are strongly determined by the properties of the waste, which are related to the quality of the combusted coal.

## 2. Materials and Methods 

### 2.1. Synthesis of Sulfur–Organic Copolymers

Sulfur organic copolymers were synthesized using a laboratory reactor (Figure 1). Sulfur (Siarkopol, Tarnobrzeg, Poland) was melted down and heated to 125–130 °C. Afterwards, an organic comonomer was added with continuous stirring. The comonomers used to synthesize the sulfur–organic copolymers were dicyclopentadiene (DCPD), styrene, 1-decene, turpentine, and furfural, all purchased from Sigma-Aldrich Co. LLC, US (Saint Louis, MO, USA). The substrates were heated to 145 °C over 90 min and then cooled to 130–135 °C, over another 90 min. The procedure for synthesis (Figure 2) has been described in our previous work [14]. The product was stirred at 130–135 °C for 180 min and then poured into flat silicone molds. The sulfur–organic copolymers consisted of 90 wt. % of sulfur, 5 wt. % of DCPD and 5 wt. % of organic monomers—styrene (SDS), turpentine (SDT), 1-decene (SDD), and furfural (SDF) (Table 1).

### 2.2. Concrete Composites

Sulfur–concrete composites based on sulfur–organic copolymers filled with aggregates and additives were prepared using a VSM-C Labmix low-speed internal mixer (Glass GmbH, Paderborn, Germany), operating with a rotational speed of 20 rpm at 135–145 °C, for about 1 hr. The aggregate fillers were—normalized granulated sand (fraction 2–8 mm) and gravel. The additives were industrial wastes, fly ash (PGE, Bełchatów, Poland), or phosphogypsum (Police S.A., Police, Poland) (Table 2). 

A fine aggregate heated to a temperature of 130 °C (266 °F) was gradually added to the liquidized binder, resulting in a drop in temperature to 125 °C (257 °F). The additives (fly ash, phosphogypsum) were then dosed before the heated coarse aggregate was added. The mixture was stirred at a temperature of at least 120 °C (248 °F), until a homogeneous consistency was achieved. Finally, it was cast in steel molds heated to 130 °C (266 °F). The sulfur concrete specimens were ready after 24 h of cooling at room temperature. Each full load portion of the mixer produced sixteen cubic specimens (of 100 mm (4 in) side length) and twelve cuboid specimens (40 × 40 × 160 mm) (1.5 in × 1.5 in × 6.25 in). These were then tested for their mechanical and physico-chemical characteristics. The main criteria for selecting the aggregates and additives were the mechanical parameters of the sulfur–concrete product. Economic considerations were of secondary importance.

### 2.3. Microorganisms

Microorganisms known for their corrosive activity against concretes were selected from the American Type Culture Collection (ATCC). The sulfur-oxidizing bacteria *Acidithiobacillus ferroxidans* ATCC 23270, *Acidithiobacillus thiooxidans* ATCC 19377 and *Thiobacillus thioparus* ATCC 8158, as well as the molds *Penicillium chrysogenum* ATCC 60739, *Aspergillus versicolor* ATCC 9577, and *Cladosporium herbarum* ATCC 6506, were used.

### 2.4. Testing the Sulfur–Organic Copolymers

#### 2.4.1. The Softening Temperature

The softening temperature of the materials was determined using a Zwick-Roell HDT/Vicat (Zwick-Roell Group, Ulm, Germany) instrument, according to ISO 306 and ASTM D 1525. Discs 3 mm high and 10 mm in diameter were placed in an oil bath, where they were subjected to 10 N perpendicular indentation. The samples were heated at a rate of dT/dt = 50 deg/h, from room temperature. The temperature reflecting 1 mm penetration of the indenter into the sample was registered as the softening point of the material.

#### 2.4.2. The Thermal Stability

The thermal stability of the materials was studied by thermogravimetric analysis (TGA). Tests were performed using a Netzsch TG 209 (Netzsch, Selb, Germany) instrument, equipped with a TASC 414/3A controller (Netzsch, Selb, Germany). Samples of around 9–10 mg were tested in a temperature range from 40 to 500 °C, with a heating rate of 10 deg/min, in air atmosphere, and with a gas flow rate of 25 cm^3^/min.

#### 2.4.3. The Melting Point of the Crystalline Phase

The melting point of the crystalline phase of the materials was determined from calorimetric analyses. Experiments were performed using a differential scanning calorimeter Netzsch DSC 204 (Netzsch, Selb, Germany) instrument, equipped with a TASC 414/3A controller (Netzsch, Selb, Germany). Samples of about 9–10 mg were tested in a temperature range from −100 to 160 °C, and with a heating rate of 5 deg/min, in air atmosphere and with a gas flow rate of 15 cm^3^/min. The amount of recrystallized sulfur in the copolymers was analyzed by wide-angle X-ray scattering (WAXS). Tests were carried out using an Empyrean System, equipped with a Cu lamp and Johansson monochromator (Malvern Panalytical Ltd., Malvern, UK). The experimental procedure used corundum as an internal standard. Each sample was enriched with an added 20% of corundum powder. The Rietveld procedure was used to calculate the content of each phase-amorphous and crystalline. The amorphous phase content was calculated as the complementary part to 100%.

#### 2.4.4. The Shore D Hardness

The shore D hardness of the materials was measured using a Zwick/Roell (Zwick-Roell Group, Ulm, Germany) instrument, according to ISO 4649 [27]. The compression strength and flexural strength of the materials were determined using a Zwick-Roell 1435 (Zwick-Roell Group, Ulm, Germany) universal mechanical testing machine, according to ISO 57 and ISO 87 standards, respectively [28,29]. The impact resistance of the materials was determined using a HIT instrument (Zwick-Roell Group, Ulm, Germany), according to ISO 179-1 [30]. Unnotched specimens were tested with a 2 J Charpy hammer operating with an angle of 170°. The stability of the exploitation characteristics of the sulfur-organic copolymers was verified by thermal aging in air at 70 °C, during 72 h. Changes to the mechanical properties of the materials caused by the aging thermal treatment were determined using the experimental procedures described above.

### 2.5. Testing of Concrete Composites

#### 2.5.1. The Compressive Strength

The compressive strength of the sulfur concrete was determined using cubic specimens (10 × 10 × 10 cm (4 in)) in accordance with the PN-EN 12390-3:2011 Standard [31]. Compressive tests were performed between 2 and 4 days, after molding. In order to determine the effect of curing time on the compressive strength, the tests were repeated 28 days after casting. The tests were conducted according to the EN 196-1 Standard [32] on cuboid specimens (40 × 40 × 160 mm (1.5 in × 1.5 in × 6.25 in)). 

#### 2.5.2. Freeze–Thaw Resistance

Freeze–thaw resistance tests were conducted in accordance to the PN-88 B-06250 Standard [33] on cubic specimens of 10 × 10 × 10 cm (4 in). The freeze–thaw resistance was determined according to two test methods—first one based on the appearance of cracks on the external concrete surface, and the second one on internal volumetric compressive strength.

In the first test method, cubic specimens from each batch were divided into two series—a reference series, kept at a room temperature of 20 ± 2 °C (68 °F) with relative humidity of 35 ± 5%, and a second series subjected to freeze–thaw cycles in air at a temperature of −20 °C (−4 °F), and in water at a temperature of +20 °C (68 °F). Each cycle took 6 h (4 h of freezing and 2 h for thawing). Freeze–thaw resistance was defined as the integrity of the external surface of the specimen after the freeze–thaw cycles (without any scratches or cracks), with a mean compressive strength of not less than 80% of the mean strength of the reference samples. 

The second type of test determines resistance to the alternating freezing and thawing in the presence of a deicing agent, and was conducted according to the EN 1338 Standard [34]. The amount of scaling per unit surface area was measured over a number of well-defined freezing and thawing cycles, under exposure to a deicing salt, mainly sodium chloride (CDF), in solution. Cubic specimens were selected and cut in half, creating two test samples from each original cube. The samples were then placed in styrofoam molds and covered with a 3% NaCl solution to a depth of 5 mm (0.25 in). The specimens were placed in a conditioner chamber and subjected to freeze–thaw cycles. The freeze–thaw cycle test was planned for 12 h, starting at a temperature of +20 °C (68 °F) for 4 h, after which the temperature was allowed to decrease with a constant cooling rate. The specimens were kept at a constant temperature of −20 °C for 3 h and then the temperature was increased with a constant heating rate over 4 h to +20 °C, where it was maintained for 1 h.

#### 2.5.3. Abrasion Resistance

Abrasion resistance was studied using two different methods—using a wide wheel and using a Böhme abrasion wheel, according to the EN 1338, EN 1339, and EN 1340 Standards [34,35,36]. Abrasion on the wide disc was evaluated by comparing the width of the groove produced on the test sample to that measured for a calibration sample. The specimen was placed on a trolley and subjected to abrasion by the corundum powder on the rotating wheel. The machine performed 75 rotations in 60 ± 3 s and automatically shut down. This test was performed on samples with minimum dimensions of 100 × 70 × 60 mm (4 in × 2.75 in × 2.375 in). To achieve reliable results, dry specimens were cleaned and the tested surfaces were painted black. This made measuring the dimensions of the resultant grooves easier and more accurate. The width of the groove was measured at 3 points with 0.1 mm accuracy. 

Abrasion resistance tests using the Böhme abrasion wheel were conducted on cubic specimens with side lengths of 7.0 ± 1.5 mm (2.75 in). A total of 20 g of abrasive was applied to abrasive belt. The abrasive disc performed 22 turns in one cycle. The whole test consisted of 16 cycles. Abrasion was defined as the volume loss (ΔV) from the test sample calculated on the basis of the previously determined density (ρR) and the mass loss after 16 cycles (Δm). Another portion of the powder was applied and the cube was mounted again in a diverted position. The weight of the cleaned sample was measured every 4 cycles and after the final cycle.

### 2.6. Testing of Biocorrosion

#### 2.6.1. The Growth of Microorganisms on the Sulfur–Organic Copolymers

The growth of microorganisms on the sulfur–organic copolymers was assessed. Molds were chosen because they are the main microorganisms that cause biodeterioration of polymeric materials. The aim was to select the most mold-resistant copolymer samples for the next stage of the study. Mold activation was performed on a Malt Extract Agar (MEA) medium (MERCK, Darmstadt, Germany) for 5 days at 27 ± 2 °C. Mold suspensions were then prepared, with a density of 10^6^ CFU/mL in a Mo1 mineral medium (NaNO_3_ 2 g, KH_2_PO_4_ 0.7 g, K_2_HPO_4_ 0.3 g, KCl 0.5 g, MgSO_4_ × 7H_2_O 0.01 g, water 1 L, pH = 5.6). The density of the suspensions was determined by microscopy (Olympus CX40 microscope, Olympus Co., Tokyo, Japan). An inoculum mixture of mold suspensions was prepared in an equal volume. The impact of the molds on the polymers was investigated in accordance with the standard (PN-EN ISO 846 2002 [37]) for liquid and solid Mo media (addition of agar 20 g) in two variants—with the addition of glucose (30 g) as a carbon source and without the addition of glucose. Polymers (discs 20 mm in diameter and 2 mm thick) were laid on the surface of the Mo solid media, which was then inoculated with 1 ml of the molds. The samples were incubated for 2 weeks at a temperature of 27 ± 2 °C, RH = 80% in a climate chamber (BINDER GmbH, Tuttlingen, Germany). The surface morphology of the polymers was then evaluated. Mold cultures were also grown in the Mo liquid with the polymers, to which 1 mL of the mold inoculation mixture was added, prior to incubation for 14 days, at temperature 27 ± 2 °C. The control samples were mixtures of molds grown in media, without polymers. After incubation, the material was filtered and the dry mass of mycelium was determined using the weight method, with a MAC 110/NH moisture analyzer (Radwag, Radom, Poland).

#### 2.6.2. The Growth of Microorganisms on the Concrete Composites

The growth of microorganisms, molds, and bacteria on three composites containing sulfur polymers was assessed. Samples of the sulfur polymer concrete composites were inoculated with 1 mL of a mold mixture, prepared as described above, and incubated for 2 months in a BINDER climatic chamber (temp. 27 ± 2 °C, RH = 80%). After 1 and 2 months, the growth of the molds was assessed based on measurement of their ergosterol content (ergosterol is one of the main components of fungal cell membranes). Extraction of the ergosterol was carried out in accordance with the methodology developed for building materials [38]. The ergosterol concentration in the samples was determined by GLC-MS, using a gas chromatograph (Agilent 7890A, USA) coupled to a mass spectrometer (Agilent MSD 5975C, USA) [28]. Samples of the sulfur concrete were also inoculated with 1 mL of a suspension of individual bacterial strains, in liquid media. The composition of the suspension depended on the species: *Acidithiobacillus thiooxidans*—(NH_4_)_2_SO_4_ 0.2 g, KH_2_PO_4_ 3 g, MgSO_4_ × 7H_2_O 0.5 g, CaCl_2_ × 6H_2_O 0.025 g, FeSO_4_ × 7H_2_O 0.01 g, sulfur 10 g, water 1 L, pH = 4.0); *Acidithiobacillus ferroxidans*—(NH_4_)_2_SO_4_ 3 g, KCl 0.1 g, K_2_HPO_4_ 0.5 g, MgSO_4_ × 7H_2_O 0.5 g, Ca(NO_3_)_2_ 0.01 g, FeSO_4_ × 7H_2_O 44.2 g, H_2_SO_4_ 1N 10 mL, water 1L, pH = 2.3); and *Thiobacillus thioparus*—(Na_2_HPO_4_ 1.2 g, KH_2_PO_4_ 1.8 g, MgSO_4_ × 7H_2_O 0.1 g, (NH_4_)_2_SO_4_ 0.1 g, CaCl_2_ 0.03 g, FeCl_3_ 0.02 g, MnSO_4_ 0.2 g, Na_2_S_2_O_3_ 10 g, water 1 L, pH = 6.6). The cultures were cultivated for 2 months at a temperature of 27 ± 2 °C, using a Unimax 1010 shaker (Haidolph, Schwabach, Germany) at 125 rpm. Bacterial growth was controlled by measuring metabolic activity based on the ATP concentration, using a HY-LITE 2 luminometer (MERCK, Darmstadt, Germany). After 1 and 2 months, changes in the morphology and chemical structure of the composite samples were assessed, and the pH of the culture fluid was measured.

#### 2.6.3. The Acid Activity Measurement

The acid activity of the molds on the sulfur polymer concrete composites was investigated in a liquid culture, using an Mo2 medium (KH_2_PO_4_ 1.0 g, (NH_4_)SO_4_ 0.3 g, MgSO_4_ × 7H_2_O 5.0 g, glucose 5 g, yeast extract 10.0 g, water 1 L), with the addition of sulfur concrete composites under the following conditions—1 mL of mold inoculum, incubation time 2 months, temperature of 27 ± 2 °C. After 1 and 2 months, the pH of the medium was measured and the samples were tested for the presence of acid metabolites produced by the molds. The pH of the liquid media, with and without the addition of sulfur polymer concrete composites, was measured using an CP-411 pH meter (ELMETRON, Zabrze, Poland). The acid activity was determined using a 10-fold concentration of the medium after the mold culture. The tests were carried out by HPLC–MS (ThermoScientific, Waltham, MA, USA), using a Surveyor liquid chromatograph with ThermoScientific PDA and RI detectors and a mass spectrometer (LTQ Velos, ThermoScientific, Waltham, Massachusetts, USA), in accordance with a previously method described [39]. 

#### 2.6.4. Morphological and Chemical Analysis of the Copolymers and Concrete Composites

Morphological analysis of the copolymers and sulfur polymer concrete composites after the cultivation of microorganisms was performed using a light microscope (OLYMPUS CX40, Olympus Co., Tokyo, Japan), at 100 × magnification, with an additional source of reflected light. Comparative analyses of the chemical structure were made by FTIR using a Fourier Transform Infrared Spectroscope (Nicolet 6700, ThermoScientific, Waltham, MA, USA), with an ATR (Attenuated Total Reflectance) for rapid analysis.

## 3. Results and Discussion

### 3.1. Characteristics of the Sulfur–Organic Copolymers

The thermal parameters of the copolymers are presented in Table 3. Partial replacement of DCPD with other comonomers shifted the softening point of the copolymers to higher temperatures. This effect was most pronounced for furfural, and least pronounced for the samples containing styrene or 1-decene. The thermal stability of the copolymers in which DCPD had been partially replaced with other organic copolymers remained almost constant, no matter the extent of the modification. However, the effect of modification was generally positive. The copolymers became more thermally stable, especially when styrene, furfural, or furfural alcohol was added.

DSC studies of sulfur–organic copolymers revealed structural changes in the copolymers due to their repeated melting. Except for 2 SDS, melting lowered the melting enthalpy of the materials (Appendix A).

Copolymerization of sulfur with organic monomers decreased its ability to recrystallize by almost half. This effect was most pronounced when pure sulfur was repeatedly melted, whereas it was hardly noticeable in the case of sulfur–styrene–DCPD copolymers (sample number 5). Partial replacement of DCPD with other monomers, especially turpentine or furfural, resulted in significant decreases in the melting enthalpy of the material. 

The degree of crystallinity can be used as a measure of the efficiency and stability of copolymerization in sulfur–organic copolymers, since it reflects the extent and durability of sulfur bonding with particular comonomers. The only component able to crystallize is “free” sulfur, which does not take part in the copolymerization reaction. Table 4 presents the crystallinity of selected, representative sulfur–organic copolymers, as determined by WAXS, and the effect of thermal ageing, on their stability (air/72 h/70 °C). 

DCPD is the basic component in sulfur–organic copolymers and is indispensable for ensuring the mechanical strength of the materials. At the same time, it is responsible for stiffness and possibly fragility. Partial replacement of DCPD makes the copolymers more resistant to compression and improves their thermal stability, while turpentine, 1-decene or furfural makes them less stiff. 

Given the importance of the thermal “history” of the materials, the mechanical and dynamical parameters of the sulfur–organic copolymers were determined, before and after thermal treatment (air/72 h/70 °C). The influence of thermal aging on the performance of the materials is presented in Table 4.

The mechanical and dynamical properties of the sulfur–organic copolymers changed significantly due to their thermal ageing of the materials. These changes were in agreement with the results for structural modification of the copolymers. The influence of changes in the degree of crystallinity, a function result of thermal aging, on the mechanical and dynamical properties of the materials was confirmed (Table 4). Sulfur–organic copolymers based on the addition of DCPD or DCPD and the studied organic monomers, became significantly stiffer after repeated melting. However, under dynamic conditions, partial replacement of DCPD with 1-decene, turpentine, or furfural lessened this effect (see Table 4—impact resistance). The addition of these compounds might be favourable from the point of view of broadening the applicability of the materials, and moreover promoted utilization of turpentine and furfural—chemicals from renewable resources. 

### 3.2. Resistance of Sulfur–Organic Copolymers to Microbial Growth

No mold growth on the tested sulfur copolymers was observed during cultivation with Mo liquid medium, either with or without the addition of glucose, nor on the solid medium without glucose. This shows that sulfur copolymers are not a source of carbon for molds. Only with the use of solid Mo medium with glucose was mycelium observed on the sulfur copolymers, in places that had been inoculated with molds. In the cases of polymer samples number 2–4, there were changes in the mycelium color and limited production of spores, compared to the control (mycelium on Mo solid). In the case of sample number 5, the mold mycelium covered the medium and the copolymer (Appendix A).

Measurements of the dry mass of the mold mycelia on liquid Mo media containing polymers, both with and without glucose, showed the highest biomass in the case of the medium with SDT copolymer (number 3) and the lowest on the medium with SDD copolymer (number 4). However, the differences in the dry masses of the mycelium were not statistically significant, which indicates that the polymers neither provided a carbon source for these microorganisms, nor released substances into the medium that inhibited mold growth (Appendix A).

Polymers number 2–4 showed no morphological changes after incubation with molds. Slight sulfur crystallization was observed only on the surface of polymer number 5, incubated with molds (Figure 3).

The FTIR spectra of copolymers number 2–4, before mold incubation, and after 2 weeks of incubation in a liquid medium, without a carbon source confirmed that no changes had occurred in their chemical structure, in terms of the distribution of absorption bands and the shapes of the peaks. In contrast, the spectra for copolymer number 5, after incubation with molds with a carbon source, showed changes in the 2400–3600 cm^−1^ wave range. These differences also occurred on the solid medium, after incubation with molds (800–2400 cm^−1^) (Figure 4).

The results indicated that copolymers number 2–4 are the most resistant to molds and can be used as raw materials for the production of sulfur concrete. In the case of copolymer number 5, mold growth led to structural changes. Therefore, in the next stage, two variants (number 2–3) of the sulfur copolymers were used. Sample number 4, based on copolymer 5 (SDF), was used for the purposes of comparison (Table 2), while sample number 1 was used as a control.

### 3.3. Characteristics of ConcreteCcomposites—Mechanical Properties and Durability

The use of additives, such as dust and good quality aggregates, can ensure that the product does not absorb water and has a high strength. Dust additives (ash, phosphogypsum) improve tightness and, thereby, reduce water absorption. The addition of fillers to sulfur concrete does not have any negative effects on the final product. The use of 2–8 mm aggregate (number 10—0.375 in) had a strong effect on the mean cube compressive strength of the sulfur concrete (Table 5). 

The mean cube compressive strength of the sulfur mortar, consisting of elementary sulfur with sand and fly ash or phosphogypsum, ranged from 49.6 MPa to 57.0 MPa. Modification of the sulfur with styrene (SDS) significantly increased the compressive strength of the sulfur mortar in samples 2–3, in comparison to mortar based on elementary sulfur, to 46.5 MPa. In the case of sample number 4 with furfural (SDF), the increase in compressive strength was not as high. In comparison to the mortar based on elementary sulfur, the difference in mean compressive strength was around 20%. With some exceptions, the strength of the sulfur concretes was significantly higher than that of the mortars. The test results indicate that the compressive strength of sulfur-concrete, based on a binder modified with dicyclopentadiene and furfural (number 4), was lower than that for sulfur concrete modified with dicyclopentadiene or styrene (number 2–3). As expected, the curing time of the sulfur concrete did not significantly affect the compressive and bending tensile strength. Full compressive strength was reached in the first 2–4 days, while 90% strength was reached during the first 24 h from casting. Only slight increases in compressive strength (compared to the early strength) were observed over 28 days.

The elasticity modulus of the sulfur concrete was higher than that of C50/60 class cement concrete (37 GPa (5,500,000 psi)).

This study also set out to determine the freeze–thaw resistance of the sulfur concrete. Concrete members for road infrastructure generally require freeze–thaw resistance of F150, corresponding to 150 cycles of freezing and thawing in water. After the test, the decrease in the strength of the frozen samples should not exceed 20% in comparison to the reference samples. Our results indicated a 35%–46% decrease in compressive strength, in comparison to the reference samples (Table 6). One reason for this low freeze–thaw resistance was the effect of soaking in water. Water penetrating into the sample weakened the interface joints at the binder. Even though the sulfur concrete was characterized by relatively low water absorption, the polymer had a decisive impact, causing a low freeze–thaw resistance. Similar freeze–thaw tests performed on the sulfur polymer confirmed the low freeze–thaw resistance of this material.

The surface scaling method was the second method used to determine freeze–thaw resistance, by measuring surface peeling after exposure to NaCl deicing salt water. This test confirmed the high freeze–thaw surface resistance of the sulfur concrete. There was negligible surface peeling after 28 freeze–thaw cycles, and only 0.3 kg/m^2^ (0.0614 lb/ft^2^), after 56 cycles. Concrete is considered to be freeze–thaw resistant to deicing salts if the total amount of surface peeling after 56 cycles is less than 1 kg/m^2^ (0.2048 lb/ft^2^) (Table 6). Given the low absorption of the binder and low freeze–thaw resistance, it might be supposed that the modifiers affected the process of sulfur binder crystallization. Crystal structures were clearly visible on some cross-section planes of the samples after failure. The addition of expanded clay aggregate led to high porosity and a more than 20% lower compressive strength (after 150 freeze–thaw cycles).

Abrasion resistance is an extremely important parameter determining the durability and long-term requirements of the sulfur concrete product. Our results confirmed that the different sulfur concrete compositions had very good resistance to abrasion, meeting the highest, Class 4, requirements (according to EN 1338, EN 1339, EN 1340 [34,35,36]) (Table 6).

Sulfur concrete is composed of sulfur as the binder, general aggregates (sand, gravel) and additives. All kinds of wastes can be used to make this material, irrespective of their quality. Compared to general concrete based on cement, sulfur concrete exhibits relatively high compressive and tensile strengths and a high corrosion resistance. Unlike general concrete cement (which reaches full concrete strength after 28 days), the final compressive and tensile strength of sulfur can be obtained a short time after solidification.

### 3.4. Resistance of Concrete Composites to Microbial Corrosion

Due to the different environments in which sulfur concrete might be used, an important feature is resistance to biocorrosion. Observations of bacterial growth by *Acidithiobacillus ferroxidans, Acidithiobacillus thiooxidans*, and *Thiobacillus thioparus*, on the sulfur concrete samples after 1 month of incubation, showed a slight color change towards yellow for some samples (Appendix A). No changes were observed in the case of molds (*Penicillium chrysogenum*, *Aspergillus versicolor*, *Cladosporium herbarum*).

Microscopic observations confirmed the recrystallization and deposition of sulfur on concrete number 4, after two months (Appendix A). The smallest changes were observed on composite number 2. The largest changes occurred on composites numbers 3 and 4, after the growth of *Acidithiobacillus ferroxidans*. This bacterium is known for its corrosive properties, producing sulfuric acid and oxidizing iron [26].

The FTIR spectra for concrete composite number 2 did not show any changes before and after 1 and 2 months of incubation with bacteria in a liquid medium (Figure 5). There were changes in the chemical structure (peaks in the ranges 2,900cm^−1^ after 1 month and about 1,000cm^−1^ after 2 months) for samples 3 and 4, with both *A. ferroxidans* and *A. thiooxidans* bacteria. However, these changes were not significant.

The growth of bacteria on media containing composites was measured on the basis of metabolic activity, indicated by the ATP content (Table 7). The ATP (adenosine triphosphate) content in the media with sulfur concrete samples inoculated with *Acidithiobacillus ferroxidans* was at the same level as that in the control sample, which suggests a lack of or very low metabolic activity in the case of this strain. In the cases of *Acidithiobacillus thiooxidans* and *Thiobacillus thioparus*, the ATP content was high, in the range of 320–13,500 Relative Light Unit (RLU). The highest concentration of ATP was measured in media with composites numbers 3 and 4. The ATP content increased with the incubation time, which might indicate favorable conditions for the growth of these two bacteria. The lowest concentration of ATP was measured in the medium with composite number 2. In this case, the ATP level decreased after 2 months for all tested bacteria, which might indicate unfavorable conditions on the surface of this sulfur concrete.

The growth of molds on the sulfur concrete samples was measured on the basis of the content of ergosterol (Table 8). After 4 weeks, the concentration of ergosterol on all samples was very high (503–798 μg/sample), indicating mold growth [38]. However, after consumption of the carbon sources (glucose and yeast extract), the content of ergosterol decreased significantly (11–17 μg/sample). This reduction in ergosterol content was due to mycelial death and degradation processes [39]. The results for ergosterol content indicate that the conditions for mold growth on the surface of the sulfur concretes were unfavorable after 60 days.

The acid activity of the bacteria and molds on the media with the addition of sulfur concrete samples was also tested (Table 9). There were no significant changes in the pH values of the media with sulfur concrete during incubation with bacteria, although a slight increase in pH over the 8-week incubation period was observed. Only in the case of sample number 3 with *T. thioparus* did the pH decrease, from 6.23 to 5.28. This bacterium is the first colonizer of concretes. It produced polythionic acid and sulfuric acid, which caused pH decreases, enabling the growth of further sulfur bacteria. This was particularly important due to the fact that metabolites biosynthesized by *Acidithiobacillus thiooxidans* have the greatest impact in the biocorrosion process [40]. These bacteria might inhabit the surface layers of concrete (1–5 mm) and cause significant reductions in pH to level 1, resulting in reactions with concrete components [21,41]. The results of our 2-month study suggest that, in this environment, the activity of biocorrosive bacteria might take longer to start, when using the studied sulfur concretes.

Acidification of the environment was observed on all media with molds. The pH decreased from 4.56–5.40 to 2.09–3.57 (Table 9). Analysis of organic acids revealed the presence of organic acids, gluconic acid, and succinic acid, in all samples of mold culture media after 60 days of incubation. In sample number 3 after 60 days, oxalic acid, citric acid, and malic acid were also identified. The presence of organic acids might result from the presence of an additional carbon source in the medium in the first stage of growth, as was similarly evidenced by a dramatic increase in the content of ergosterol. The presence of these organic acids in model studies on media with building materials inoculated with molds, had already been confirmed in previous studies [42]. The production of both inorganic and organic acids led to an increase in the solubility of concrete and mortar, due to the formation of chelated compounds with metal cations (calcium, aluminum, silicon, iron, manganese, magnesium) or water-soluble salts [42]. The long-term effects of microorganisms and their metabolites might cause material damage, such as crushing and abrasion [43,44]. Microbial corrosion of concrete occurs during contact with sewage, agricultural waste, or soil, in which microbial contamination is high. In the case of molds, the presence of organic compounds on the surface of the sulfur concrete was confirmed.

## 4. Conclusions

In this study, sulfur–organic copolymers containing 90 wt.% of sulfur and 5 wt.% of DCPD were produced, completed with 5 wt.% of an organic monomer—styrene (SDS), turpentine (SDT), 1-decene (SDD), or furfural (SDF). The polymeric materials were found to be chemically stable. Partial replacement of DCPD with other organic comonomers did not change the thermal stability markedly but did make the copolymers more elastic. However, the materials became significantly stiffer after repeated melting. Partial replacement of DCP with 1-decene, turpentine, or furfural reduced this effect under dynamic conditions. All the tested copolymers were found to be resistant to microbial corrosion. The highest resistance to mold growth was noted for the SDS-containing polymer. Polymers containing SDT and SDD showed less resistance, while the SDF polymer underwent the greatest changes (in terms of FTIR spectra and sulfur crystallization) after incubation with microorganisms. Sulfur–organic copolymers containing styrene can be used as engineering materials or applied as binders in sulfur-concretes.

The concrete composites with sulfur–organic copolymers containing DCPD, SDS, SDF, fly ash, and phosphogypsum were mechanically resistant to compression and stretching, had low water absorbance, and were resistant to factors such as temperature and salt. Resistance to freezing and thawing (150 cycles) was not confirmed.

The concrete composites with sulfur–organic copolymers showed resistance to bacterial growth and acid activity during 8 weeks of incubation with microorganisms. No significant structural changes were observed in the SDS composites after incubation with bacteria, whereas composites containing SDF showed slight changes (FTIR, microscopic analysis). The concrete composite containing sulfur, DCPD, SDS, sand, gravel, and fly ash was the most resistant to microbiological corrosion, based on the metabolic activity of the bacteria and the production of ergosterol by the molds, after 8 weeks of incubation. It was found that *T. thioparus* was the first of the acidifying bacteria to colonize the sulfur concrete, decreasing the pH of the environment. The molds *P. chrysogenum*, *A. versicolor*, and *C. herbarum* were able to grow on the surface of the tested composites only in the presence of an organic carbon source (glucose). During incubation, they produced organic acids and acidified the environment. However, no morphological changes in the concretes were observed. Such changes might occur in the case of contact with sewage, agricultural waste, or soil, over extended periods. 

In summary, the ecological concrete composites are suitable for use in the construction of internal elements, such as small prefabricated elements, due to their good strength parameters and resistance to external factors. However, their use is not recommended under conditions of external exposure to changing atmospheric parameters (freezing and thawing cycles) or contact with sewage and soil (with high content of microorganisms and organic compounds, stimulating microbial metabolic activity).

## 5. Patents

Kotynia, R.; Wlendziak, R.; Pawlica, J.; Berlowska, J.; Dziugan, P.; Palka, K.; Tynenski Z. Application in Polish Patent Office No P.422745 “Sulfur concrete mix”.

Gozdek, T.; Imiela, M.; Sicinski, M. Tynenski Z.; Palka, K.; Bielinski, D.; Anyszka, R. Application in Polish Patent Office No P.422745 “Method for obtaining stable polymeric sulfur that is a binder for sulfur concretes and the stable polymeric sulfur that is a binder for sulfur concretes”.

## Figures and Tables

**Figure 1 materials-12-02602-f001:**
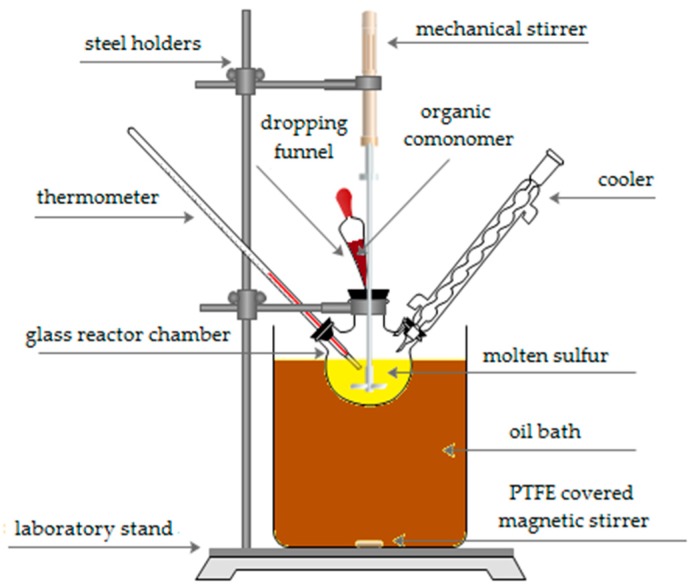
Laboratory equipment used for synthesis of sulfur–organic copolymers.

**Figure 2 materials-12-02602-f002:**
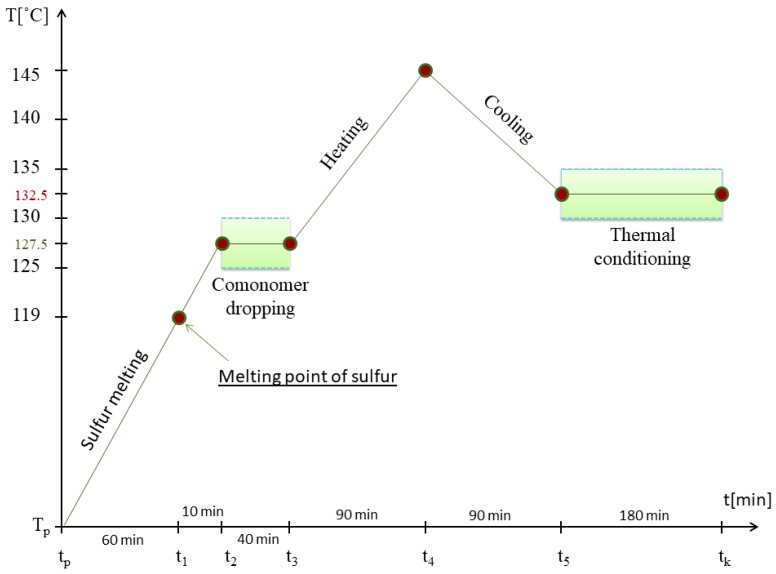
Procedure used in the synthesis of sulfur–organic copolymers.

**Figure 3 materials-12-02602-f003:**
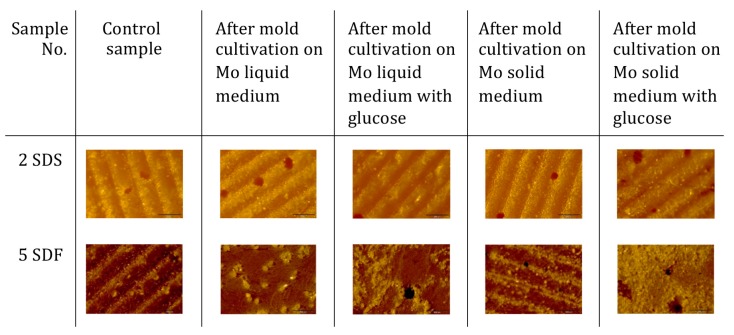
Microscopic observations of selected sulfur copolymers after incubation with molds.

**Figure 4 materials-12-02602-f004:**
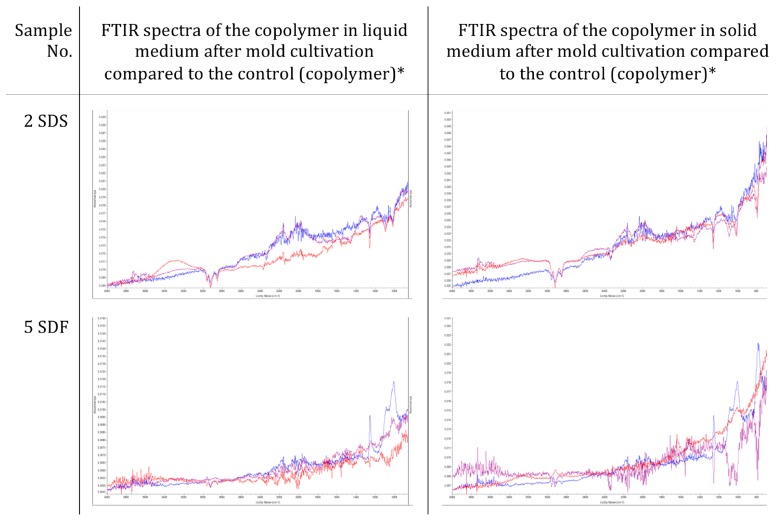
FTIR spectra of the selected sulfur copolymers after incubation with molds (blue—copolymer before incubation, purple—after 14 days of mold cultivation in the medium without carbon source, red—after 14 days of molds culture in the medium with a carbon source).

**Figure 5 materials-12-02602-f005:**
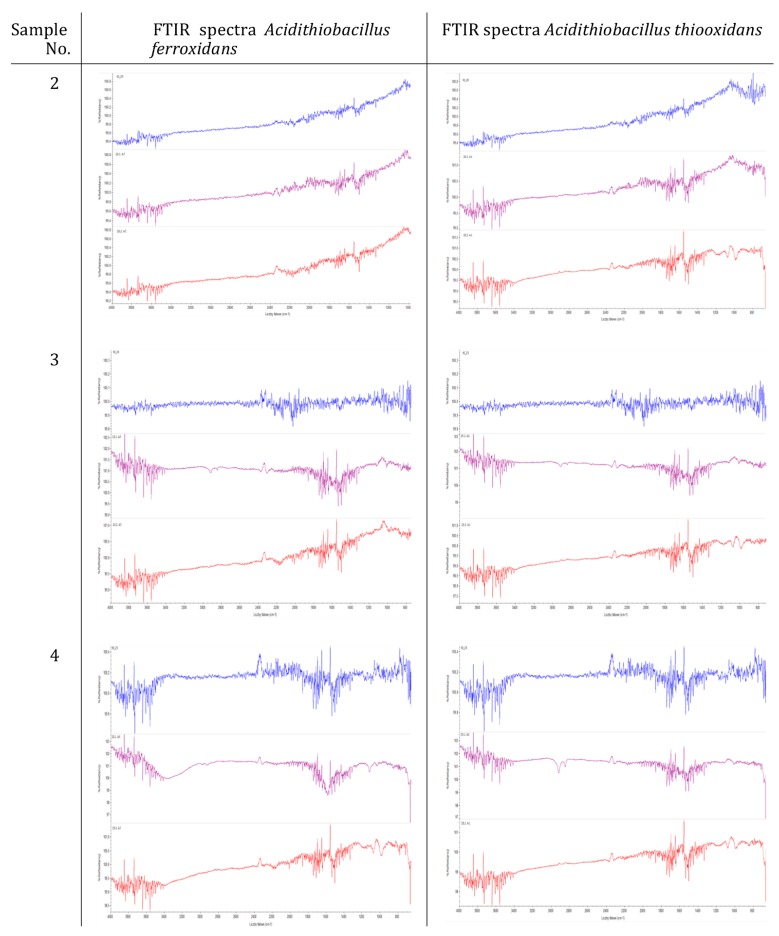
FTIR spectra of sulfur concrete after incubation with bacteria (blue—sulfur concrete before incubation, purple—after 1 month of incubation with bacteria, red—after 2 months of incubation with bacteria).

**Table 1 materials-12-02602-t001:** Composition of the synthetized sulfur–organic copolymers.

Sample Number	Content (%)			
	DCPD	Sulfur	Organic Additives	
1 control	-	100	-	-
2 SDS	5	90	styrene	5
3 SDT	5	90	turpentine	5
4 SDD	5	90	1-decene	5
5 SDF	5	90	furfural	5

SDS‒copolymer with styrene; SDT‒copolymer with turpentine; SDD‒copolymer with 1-decene; SDF‒copolymer with furfural.

**Table 2 materials-12-02602-t002:** Composition of the sulfur–concrete composites.

Sample Number	Content (%)			
	Sulfur–Organic Copolymers		Aggregate Type	Additives Type
1 control	Sulfur	30.0	Sand 60.0	Fly ash 10.0
2	Sulfur DCPD SDS	21.1 0.55 0.55	Sand 15.6 Gravel 54.4	Fly ash 7.8
3	Sulfur DCPD SDS	24.5 0.65 0.65	Sand 6.4 Gravel 58.7	Phosphogypsum 9.1
4	Sulfur DCPD SDF	24.5 0.65 0.65	Sand 13.1 Gravel 52.0	Phosphogypsum 9.1

**Table 3 materials-12-02602-t003:** Thermal parameters of sulfur–organic copolymers.

Sample Number	Parameter				
	Softening Point (°C)	Melting Temp. Range (°C)	Melting Enthalpy (J/g)	Thermal Stability T_5_ (°C)	Thermal Stability T_50_ (°C)
2 SDS	63.0 ± 2.0	80–115	37.79	238	295
3 SDT	105.0 ± 1.0	85–120	33.77	226	285
4 SDD	75.0 ± 2.0	95–120	41.27	221	272
5 SDF	90.0 ± 1.0	95–125	47.31	220	288

**Table 4 materials-12-02602-t004:** Influence of thermal ageing (air/72 h/70 °C) on mechanical, dynamical parameters of the sulfur–organic copolymers and the degree of crystallinity in the sulfur–organic copolymers.

Sample Number	Compression Strength (N)		Flexural Strength (MPa)		Hardness (°Sh D)		Impact Resistance (J/m^2^)		S Crystal Content before Aging (wt.%)	
	virgin	after aging	virgin	after aging	virgin	after aging	virgin	after aging	virgin	after aging
1 control	204 ± 57	109 ± 76	4.4 ± 3.5	1.0 ± 0.2	27.5 ± 0.3	27.6 ± 0.3	38.2 ± 6	91.3 ± 9	52.3	61.4
2 SDS	383 ± 71	599 ± 46	4.4 ± 3.0	6.3 ± 4.0	27.0 ± 0.1	27.0 ± 0.1	63.4 ± 2	186.8 ± 16	50.0	57.4
3 SDT	399 ± 51	188 ± 32	3.7 ± 2.6	2.5 ± 0.6	27.0 ± 0.1	27.5 ± 0.2	44.3 ± 2	131.1 ± 12	63.4	50.5
4 SDD	234 ± 50	14 ± 15	5.6 ± 0.8	2.1 ± 0.7	26.9 ± 0.1	27.3 ± 0.2	35.8 ± 2	126.0 ± 10	59.5	48.4
5 SDF	381 ± 58	256 ± 61	4.1 ± 0.5	1.2 ± 0.8	27.1 ± 0.2	27.4 ± 0.2	42.2 ± 6	129.3 ± 12	66.8	69.0

**Table 5 materials-12-02602-t005:** Mechanical tests of the sulfur–concrete composites.

Sample Number	Bulk Density kg/m^3^ (pcf)	E compressive Strength Cube Samples MPa (psi)	L Compressive Strength Cube Samples MPa (psi)	Compressive Strength Cuboid Samples MPa (psi)	Tensile Bending Strength Cuboid Samples MPa (psi)
1 C	2260 (141)	46.5 (6744)	- *	38.3 (5555)	5.6 (812)
2	2390 (149)	57.0 (8267)	51.7 (7498)	66.1 (9587)	7.3 (1059)
3	2240 (140)	50.6 (7339)	68.7 (9964)	58.3 (8456)	13.3 (1929)
4	2260 (141)	49.6 (7194)	50.5 (7324)	51.6 (7484)	4.5 (653)

C—control; E—early compressive strength determined 2–4 days from production day; L—lateral compressive strength determined after 28 days from production day; * not enough specimens to carry out the tests; 1 MPa = 145.038 psi; 16.02 kg/m^3^ = 1 lb/ft^3^.

**Table 6 materials-12-02602-t006:** Durability of sulfur concrete.

Sample Number	Test									Requirements
	**Freeze–Thaw (FT) Resistance**									
	Compressive strength reference samples MPa (psi)	Compressive strength tested samples MPa (psi)			Strength loss in comparison to reference samples (%)		Number of performed cycles		Water absorption after 24 h (%)	
2	57.0 (8257)	31.2 (4525)			45		198		0.47	Strength loss in comparison to reference samples max. 20% and no visible cracks
3	56.1 (8137)	36.3 (5265)			35		211		0.07	
4	49.6 (7194)	26.9 (3902)			46		100		0.35	
	**Freeze–Thaw (FT) Resistance in 3% Salt Solution (Scaling)**									
	Mass of scaled material after 28 cycles (m_28_) (g)		Mass of scaled material after 56 cycles (m_56_) (g)			Mass of scaled material after 28 cycles (m_28_) (kg/m^2^)		Mass of scaled material after 56 cycles (m_56_) (kg/m^2^)		
1A	1.50		2.51			0.16		0.26		FT2 m_28_ ≤ 0.5 m_56_ ≤ 1.0 (kg/m^2^)
2B	0.01		0.13			0		0.01		
3B	0.03		0.11			0		0.01		
4B	0.07		0.36			0.01		0.04		
	**Böhme’s Plate Abrasion Tests**									
	Sample mass loss (g)			Sample volume loss (g/mm^3^)			Results (mm^3^/5 000 mm^2^)			
1	23.37			10 260			10 095			Class 4 ≤ 18,000
2	30.35			13 239			13 276			
	**Wide Wheel Abrasion Tests—Width of Groove**									
	Man value (mm)				Results correlated to calibration sample (mm)					
2	15.1				20.0					Class 4 ≤ 20 mm
3	14.4				19.0					
4	15.0				19.5					

1—control sample; A—surface cut with a saw, exposed aggregate; B—mold surface.

**Table 7 materials-12-02602-t007:** ATP (RLU/sample) content in media with sulfur concrete after incubation with bacteria.

Sample Number	*Acidithiobacillus ferroxidans* ^1^		*Acidithiobacillus thiooxidans* ^2^		*Thiobacillus thioparus* ^3^	
	after 30 days	after 60 days	after 30 days	after 60 days	after 30 days	after 60 days
2	X: 5.5	X: 4.0	X: 2603.5	X: 1402.0	X: 2653.5	X: 715.0
	SD: 0.7	SD: 1.4	SD: 3672.0	SD: 1977.1	SD:3742.7	SD: 148.9
3	X: 6.0	X: 5.5	X: 126.0	X: 6650.0	X: 5.0	X: 13500.0
	SD: 0.0	SD: 2.1	SD: 175.4	SD: 7566.0	SD: 1.4	SD: 2121.3
4	X:5.0	X: 4.0	X: 32.0	X: 6300.0	X: 840.0	X: 4600.0
	SD: 0.0	SD: 0.0	SD: 0.0	SD: 424.3	SD: 0.0	SD: 3676.9

ATP content in media at time t = 0 h: ^1^ 14 RLU/sample; ^2^ 5 RLU/sample; ^3^ 7 RLU/sample; concreate sample—2 × 5 × 1 cm; X—mean; SD—standard deviation.

**Table 8 materials-12-02602-t008:** Ergosterol content on sulfur concrete samples after incubation with molds.

Sample Number	Ergosterol Content (μg/sample)	
	ater 30 days	after 60 days
2	X: 502.6	X: 16.64
	SD: 45.26	SD: 0.83
3	X: 662.36	X: 12.43
	SD: 72.86	SD: 1.24
4	X: 797.98	X: 10.92
	SD: 39.90	SD: 0.87

Ergosterol content on concrete samples at time t = 0 h: 7.0 ± 0.05; mg/sample; sample—2 × 5 × 1 cm; X—mean; SD—standard deviation.

**Table 9 materials-12-02602-t009:** The pH of media with sulfur-concrete samples after incubation with microorganisms.

Sample Number	*Acidithiobacillus* *ferroxidans* ^1^		*Acidithiobacillus* *thiooxidans* ^2^		*Thiobacillus* *thioparus* ^3^		*Penicillium chrysogenum, Aspergillus versicolor, Cladosporium herbarum*	
	after 30 days	after 60 days	after 30 days	after 60 days	after 30 days	after 60 days	after 30 days	after 60 days
2	1.34 ± 0.02	1.43 ± 0.01	4.72 ± 0.11	4.74 ± 0.07	6.38 ± 0.33	6.50 ± 1.15	5.40 ± 0.30	2.09 ± 0.01
3	1.35 ± 0.18	1.52 ± 0.21	4.08 ± 0.11	4.20 ± 0.12	6.23 ± 1.44	5.28 ± 2.14	4.56 ± 0.19	2.71 ± 1.06
4	1.41 ± 0.01	1.57 ± 0.01	4.88 ± 0.01	5.08 ± 0.01	6.31 ± 0.01	6.35 ± 0.01	4.90 ± 0.10	3.57 ± 0.10

Starting pH of culture media for bacteria: ^1^ pH = 2.3 ± 0.01; ^2^ pH = 4.0 ± 0.01; ^3^ pH = 6.8 ± 0.01; starting pH of culture media for molds: 4.80 ± 0.01.

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
