# Peer review of "New Sulfur Organic Polymer-Concrete Composites Containing Waste Materials: Mechanical Characteristics and Resistance to Biocorrosion"

_materials, 2019, doi:10.3390/ma12162602_

Round 1

Reviewer 1 Report

The authors describe new sulfur organic polymer-concrete composites containing waste materials and analysed their properties. The topic is of high interest and utility furthermore the research is sustainable as it includes potential recycling of industrial waste. The present research is conducted and described well thus it deserves in my humble opinion immediate publication. 

Author Response

We would like to thank the Reviewer for a thorough evaluation of our article.

Reviewer 2 Report

This paper presents a study of developingnew sulfur-copolymer concrete composites based on waste compounds. The paper is written well. However reviewer has some minor comments:

Abstract needs to be mosified. Some numerical results need to be added.

The writing is not consistent. In many places authors used used space after degree(o) before C. Then some places space before degree and some places no space.

There should be no space between number and % (e.g., page 2 line 90)

Page 5: line 161 what is ETo?

Is it economically viable to develop new sulfur-copolymer concrete composites based on waste compounds compared to cement?

The paper is too long. It should be truncated significantly. Specially section 2.4, 2.5 and 2.6 can be merged with section 3.

What is the effect of moisture concentration on microbial corrosion? 

Rewrite the conclusion to reflect the abstract.

Author Response

We would like to thank the Reviewer for a thorough evaluation of our article and we believe that observations will eliminate any shortcomings/errors that appeared in the text.

Below there are replies to the comments:

1) Abstract needs to be mosified. Some numerical results need to be added. 

Rewrite the conclusion to reflect the abstract.

Thank Reviewer for this comment. According to the reviewer's suggestion, we have changed the abstract to reflect conclusion.

Abstract (old version): The aim of this study was to develop new sulfur-copolymer concrete composites based on waste compounds with good mechanical characteristics and resistance to biocorrosion. The comonomers used to synthesize the sulfur-organic copolymers were: 90 wt. % of sulfur; 5 wt. % of dicyclopentadiene (DCPD); 5 wt. % of organic monomers, styrene (SDS), 1-decene (SDD), turpentine (SDT) and furfural (SDF). The concrete composites based on sulfur-organic copolymers were filled with aggregates, sand and gravel, as well as additives, industrial waste such as fly ash or phosphogypsum. The mechanical and physico-chemical characteristics of the sulfur-organic copolymers (softening temperature, thermal stability, melting temperature, amount of recrystallized sulfur, shore D hardness, compression strength and flexural strength, impact resistance, stability of exploitation) and of the concrete composites (compressive strength,  freeze-thaw resistance, abrasion resistance) were determined, as well as their resistance to biocorrosion induced by sulfur-oxidizing bacteria and molds.The sulfur-organic copolymers were found to be chemically stable. However, the materials became significantly stiffer after repeated melting. All the tested copolymers were resistant to microbial corrosion. The highest resistance was exhibited by the SDS-containing polymer, while the SDF polymer exhibited the greatest change due to the activity of the microorganisms (FTIR analysis, sulfur crystallization). 

Abstract (new version):The aim of this study was to develop new sulfur-copolymer concrete composites based on waste compounds with good mechanical characteristics and resistance to biocorrosion. The comonomers used to synthesize the sulfur-organic copolymers were: 90 wt. % of sulfur; 5 wt. % of dicyclopentadiene (DCPD); 5 wt. % of organic monomers, styrene (SDS), 1-decene (SDD), turpentine (SDT) and furfural (SDF). The concrete composites based on sulfur-organic copolymers were filled with aggregates, sand and gravel, as well as additives, industrial waste such as fly ash or phosphogypsum. The sulfur-organic copolymers were found to be chemically stable (softening temperature, thermal stability, melting temperature, amount of recrystallized sulfur, shore D hardness).Partial replacement of DCPD with other organic comonomers did not change the thermal stability markedly but did make the copolymers more elastic.However, the materials became significantly stiffer after repeated melting. All the tested copolymers were found to be resistant to microbial corrosion. The highest resistance was exhibited by the SDS-containing polymer, while the SDF polymer exhibited the greatest change due to the activity of the microorganisms (FTIR analysis, sulfur crystallization). The concrete composites with sulfur-organic copolymers containing DCPD, SDS, SDF, fly ash and phosphogypsum were mechanically resistant to compression and stretching,had low water absorbance and were resistant to factors such as temperature and salt. Resistance to freezing and thawing (150 cycles) was not confirmed. The concrete composites with sulfur-organic copolymers showed resistance to bacterial growth and acid activity during 8 weeks of incubation with microorganisms. No significant structural changes were observed in the SDS composites after incubation with bacteria, whereas composites containing SDF showed slight changes (FTIR, microscopic analysis). The concrete composite containing sulfur, DCPD, SDS, sand, gravel and fly ash was the most resistant to microbiological corrosion, based on the metabolic activity of the bacteria and the production of ergosterol by the molds after 8 weeks of incubation. It was found thatT. thioparuswas the first of the acidifying bacteria to colonize the sulfur concrete, decreasing the pH of the environment. The molds P. chrysogenum,A. versicolorandC. herbarumwere able to grow on the surface of the tested composites only in the presence of an organic carbon source (glucose). During incubation, they produced organic acids and acidified the environment. However, no morphological changes in the concretes were observed. Sulfur-organic copolymers containing styrene can be used as engineering materials or applied as binders in sulfur-concretes.

2) The writing is not consistent. In many places authors used used space after degree(o) before C. Then some places space before degree and some places no space. There should be no space between number and % (e.g., page 2 line 90)

According Reviewer’s note we have changed in whole manuscripts space between number /degree and % or C.

3) Page 5: line 161 what is ETo?

It is typo mistake, Eto should be removed in sentence :  EToEach sample was enriched with added 20 % of corundum powder.

4) Is it economically viable to develop new sulfur-copolymer concrete composites based on waste compounds compared to cement?

A natural result of stockpiles growth is to reuse the recycled sulfur obtained from a purifying fuel oil process in the production of new structural products. Replacement of cement with sulfur polymer as a binder in concrete compounds, is one of direction to utilize a sulfur waste. To solve the problem of degradation of internal sulfur polymer bonds, several modifiers of standard properties of  sulfur binders can be applied. The advantage of using a modified sulfur polymer as a binder in manufacturing sulfur concrete is an ability to gain the full compressive strength very fast, already after 24 hours from forming it. Another potential benefit of using sulfur concrete is to utilize  still  growing wastes kept at stockpiles.

As it was presented in the paper (page 2, lines:86-96): According to GUS (Central Statistical Office of Poland) 131,3 million tons of industrial waste materials were produced in Poland in 2014 and only 27,6 % of them were recycled. Total mass of wastes  kept at stockpiles  reached 1683,5 million tons in 2014. Slag and fly ashes (products of coal combustion) have been the most often stored at stockpiles near power plants and electrical power and heating stations. Only in Poland there were established 48 of stockpiles. Most of those materials might be used in production of concrete but the possibility of application is strictly related to physic-chemical properties of wastes and those are related to a class of fuel (quality of combusted coal). Not every one set of the feed stock is suitable for direct usage in concrete production without previous refinement.

A sulfur concrete is a material composed of sulfur, as the binder and variable types of aggregates (sand, gravel, stone chips, granite and basalt) or light aggregates (slag sand, pumice, expanded clay and tuff) at the temperature ranged between 130 and 160°C (266 and 320°F). This material gives possibility to use all kinds of wastes regardless of their quality.

Comparing to general concrete based on cement, the sulfur concrete is as a material, which exhibits increased compressive and tensile strength and the corrosion resistance. It reaches the final compressive and flexural strength a short time after solidification not as is attained in the concrete (after 28 days). The sulfur concrete is characterized by high durability in a harsh environment such as dilute acids and to salts. It is also resistant to biodegradation, radiation activity (Szajerski, 2019) and freeze-thaw cycles. It is a hardly liquid absorbing material with a high compressive strength , easily for recycle.

These advantages of sulfur concrete give opportunity of a wide range of its applications in civil engineering, hydraulic engineering, road constructions, structures exposed to aggressive environments, chemical industry, petroleum industry, food industry and farming.

4) The paper is too long. It should be truncated significantly. Specially section 2.4, 2.5 and 2.6 can be merged with section 3.

Manuscript presents completed study, four figures and one table were included in supplementary  materials. According to Reviewer’s suggestion sections 2.4- 2.6 are truncated into subsections. Section 3. is description of the results with 4 subsections.

5) What is the effect of moisture concentration on microbial corrosion? 

The high porosity of concrete and the lack water resistance significantly affect the intensity of biological corrosion. In residential buildings, molds have been observed on concrete in conditions of high relative humidity WWP = 70-80% and temperature T = 18-22°C, despite the high pH = 8.4-9.7 (conditions as in the studies presented in manuscript), intense growth of Alternaria alternata, Aspegillus niger, Penicillium chrysogenum and Cladosporium cladosporioideswas found on the surface. Concrete surfaces that have contact with other materials (paper, textiles) should not exceed 3% moisture, in such conditions molds do not develop (Gutarowska, 2013).

Reference:

Gutarowska B. (2013) Moulds and their allergens in buildings. Ed LAP LAMBERT Academic Publishing , Germany, ISBN 978-3-659-33657-7.

Szajerski P., Celinska J., Bem H., Gasiorowski A., Anyszka R., Dziugan P. (2019) Radium content and radon exhalation rate from sulfur polymercomposites (SPC) based on mineral fillers. Construction and Building Materials 198 (2019) 390–398

We hope that our explanations and all the corrections made in the article will make it more valuable for readers of Materials.

Yours sincerely,

Authors

Round 2

Reviewer 2 Report

Authors made substantial changes to the manuscript. Papers can be accepted for publication.